# Improving Multi-Criteria Chinese Word Segmentation through Learning Sentence Representation

**Chun-Yi Lin, Ying-Jia Lin, Yi-Ting Li, Chia-Jen Yeh, Ching-Wen Yang, Hung-Yu Kao**

Department of Computer Science and Information Engineering
National Cheng Kung University
Taiwan (R.O.C)
`ne6101050@gs.ncku.edu.tw,`
`{yingjia.lin.public, yitingli.public, chiaren.yeh.public}@gmail.com,`
`p76114511@gs.ncku.edu.tw, hykao@mail.ncku.edu.tw`

## Abstract

Recent Chinese word segmentation (CWS) models have shown competitive performance with pre-trained language models' knowledge. However, these models tend to learn the segmentation knowledge through in-vocabulary words rather than understanding the meaning of the entire context. To address this issue, we introduce a context-aware approach that incorporates unsupervised sentence representation learning over different dropout masks into the multi-criteria training framework. We demonstrate that our approach reaches state-of-the-art (SoTA) performance on F1 scores for six of the nine CWS benchmark datasets and out-of-vocabulary (OOV) recalls for eight of nine. Further experiments discover that substantial improvements can be brought with various sentence representation objectives.

## 1 Introduction

Chinese word segmentation (CWS) is a fundamental step for Chinese natural language processing (NLP) tasks. Researchers have publicized various labeled datasets for evaluating CWS models. However, due to the varied properties among the CWS datasets, different segmentation criteria exist in different datasets (Chen et al., 2017; Huang et al., 2020a). A straightforward solution is to create a model for each segmentation criterion (Tian et al., 2020), but this constrains the model from learning cross-dataset segmentation instances.

In order to facilitate the differentiation of various segmentation criteria, researchers started to work on building multi-criteria CWS (MCCWS) models. Common MCCWS models employ either a single encoder with multiple decoders (Chen et al., 2017; Huang et al., 2020b) or a single model with extra special tokens (He et al., 2019; Huang et al., 2020a; Ke et al., 2020; Qiu et al., 2020; Ke et al., 2021). The former assigns distinct decoders to different criteria, sharing other model parts. The

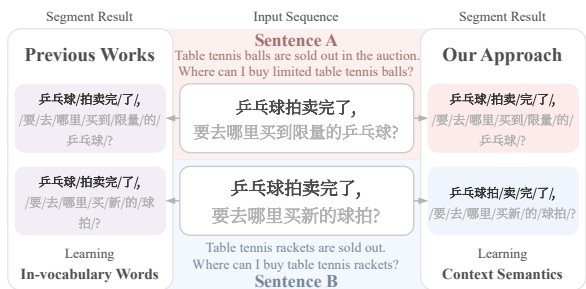

Figure 1: Comparison of previous works and ours

latter uses special tokens at each input's start, serving the same purpose as private decoders, for a compact model that still differentiates segmentation criteria. However, both of the approaches tend to overfit the in-domain majority criteria for each dataset in use and therefore fail to provide correct segmentations for the minority words, especially the context-dependent ones. We show an example in Figure 1.

In this paper, we present a context-aware approach to improve MCCWS. To enhance the model's understanding of context, inspired by Gao et al. (2021), we leverage the randomness of dropout (Srivastava et al., 2014) and introduce an auxiliary task for minimizing the difference of sentence representations under the multi-criteria training framework. Our contributions lie in (1) The proposed approach sets a new state-of-the-art in the evaluations of F1-score and OOV recall on several CWS datasets. (2) Various objective designs for sentence representation learning can also be effective for improving CWS.

## 2 Related work

After Xue (2003) first treated the CWS task as a character tagging problem, most successive researchers followed the approach and performed well. Chen et al. (2017) incorporated multiple CWS datasets by matching a specific decoder to

different criteria during training. Inspired by the idea of a decoder for each criterion, Huang et al. (2020b) used a pre-trained language model as their encoder, further enhancing CWS performance. But the cost of maintaining decoders increases when the MCCWS model addresses more datasets. To reduce the model parameters, He et al. (2019); Ke et al. (2020); Qiu et al. (2020); Huang et al. (2020a); Ke et al. (2021) utilized extra special tokens to represent criteria respectively, such that the MCCWS model can interpret the extra special token as a hint and segment text differently.

Learning sentence representation enhances the pre-trained language model's contextual understanding. In recent years, contrastive learning (Gao et al., 2021; Chuang et al., 2022; Zhang et al., 2022) has been the most popular method for learning sentence representation without additional annotated data. Maximizing the similarity of the hidden representation of the same text masked by different masks and minimizing the similarity of the hidden representation of different text help the model further understand the text. Their model performs better on natural language understanding (NLU) tasks after training on the sentence representation task. As a result, we use an extra special token as a hint to control the criterion and add a sentence representation task to produce the SoTA performance.

## 3 Methodology

### 3.1 Chinese Word Segmentation

Our model is based on pre-trained Chinese BERT (Devlin et al., 2019). Suppose that we have $M$ datasets $\{D^k\}_{k=1}^M$. Each input sentence $s$ from a dataset $D^k$ transforms into a sequence as below:

$$s = [[CLS]; [CT]; s; [SEP]], \tag{1}$$

where [CLS] and [SEP] are special tokens for the pre-trained language model, and [CT] is the criterion token for each dataset $D^k$. We denote the hidden representation of each token in index $i$ output from BERT as $h_i$. As a sequence labeling task, the MCCWS model outputs a vector $y_i$ consisting of each label's probability to each input character $s_i$. Each element in $y_i$ stands for the probability of each label in the label set $\mathcal{A} = \{B, M, E, S\}$, and B, M, E stands for the beginning, middle, and end of a word, and S represents a word that only has a single character. To accelerate the decoding process and make our model simple, we replace the

CRF (Lafferty et al., 2001) layer with a linear layer as our decoder. Our decoder can form as follows:

$$y_{i-2} = \text{softmax}(W^d \cdot h_i + b^d) \in \mathbb{R}^4,$$
$$\forall i \in \{2, ..., |s| + 1\},$$

where $W^d$ and $b^d$ are trainable parameters. We use the cross-entropy function as our loss function to force our model to predict the correct label on each character.

$$\mathcal{L}_{\text{ws}} = -\frac{1}{|s|} \sum_{i=1}^{|s|} 1 \cdot \log y_i^l, \tag{2}$$

where $y_i^l$ represents the probability of the correct label $l$ given by our model.

### 3.2 Criterion Classification

To let our model distinguish the criteria accurately, we refer to the approach proposed by Ke et al. (2020) and train the criterion token with a classification task. These criterion tokens can also be viewed as control tokens that manually prompt the model to segment sentences using different criteria. We can form the criterion classifier as follows:

$$c = \text{softmax}(W^c \cdot h_1 + b^c) \in \mathbb{R}^M. \tag{3}$$

Both $W^c$ and $b^c$ are trainable parameters. $M$ is the number of datasets we used for training. The function we used for training criterion classification is cross-entropy loss and can be formed as:

$$\mathcal{L}_c = -1 \cdot \log c^k, \tag{4}$$

where $c^k$ represents the probability given by our model of the input dataset $D^k$.

### 3.3 Learning Sentence Representation

To make our model further understand the input text, we add the sentence representation loss to our training objective. Following the contrastive learning method proposed by Gao et al. (2021), we pass every sequence s through the encoder with different dropout masks. The two hidden representations are a pair of positive samples. The pair of negative samples are combined with two hidden representation vectors from different input sentences. We pick up the two hidden representations of the same input sequence $i$ at index 0, which is the hidden representation of [CLS] token, and denote them as $h_{0i}$

| MCCWS Models | AS | CIT | CNC | CTB6 | MSR | PKU | SXU | UD | ZX | Avg.4 | Avg.6 | Avg.9 |
|---|---|---|---|---|---|---|---|---|---|---|---|---|
| Gong et al., 2019 | 95.22 | 96.22 | - | 97.62 | 97.78 | 96.15 | 97.25 | - | - | 96.34 | 96.71 | - |
| Huang et al., 2020a | - | - | 97.19 | 97.56 | 98.29 | 96.85 | 97.56 | 97.69 | 96.46 | - | - | - |
| Huang et al., 2020b | 97.00 | 97.80 | 97.30 | 97.80 | **98.50** | **97.30** | 97.50 | 97.80 | 97.10 | **97.65** | 97.65 | 97.57 |
| Qiu et al., 2020 | 96.44 | 96.91 | - | 96.99 | 98.05 | 96.41 | 97.61 | - | - | 96.95 | 97.07 | - |
| Ke et al., 2020 | 96.90 | 97.07 | - | 97.20 | 98.45 | 96.89 | 97.81 | - | - | 97.33 | 97.39 | - |
| Ke et al., 2021 | **97.04** | 98.12 | 97.25 | 97.87 | 98.02 | 96.76 | 97.51 | 83.84 | 88.48 | 97.49 | 97.55 | - |
| Ours | 96.81 | **98.16** | **97.41** | **97.93** | 98.39 | 96.94 | **97.82** | **98.30** | **97.12** | 97.58 | **97.68** | **97.65** |

Table 1: F1-score (in percentage) on all nine datasets. (We show the result of the significance test in Table 7.) Avg.4: Average among AS, CIT, MSR, and PKU; Avg.6: Average among AS, CIT, CTB6, MSR, PKU, and SXU; Avg.9: Average among all nine datasets; Ours: Our method

and $h_{0i}^+$. Then training our model by an objective for $(h_{0i}, h_{0i}^+)$ with a batch of $N$ pairs is:

$$\mathcal{L}_s = -\log \frac{e^{\text{sim}(h_{0i}, h_{0i}^+)/\tau}}{\sum_{j=1}^{N} e^{\text{sim}(h_{0i}, h_{0j}^+)/\tau}}, \quad (5)$$

where $\tau$ is a hyperparameter of temperature and $\text{sim}(h_{0i}, h_{0i}^+)$ is the cosine similarity $\frac{h_{0i}^T h_{0i}^+}{||h_{0i}|| \cdot ||h_{0i}^+||}$.

### 3.4 Total Loss

Combining Equation(2), Equation(4), and Equation(5), we get the final training objective $\mathcal{L}_{\text{total}}$:

$$\mathcal{L}_{\text{total}} = \mathcal{L}_{\text{ws}} + \alpha \cdot \mathcal{L}_c + (1 - \alpha) \cdot \mathcal{L}_s, \quad (6)$$

where $\alpha$ is a hyperparameter to control how the model weighs between criterion classification loss and the sentence representation loss.

## 4 Experiment

### 4.1 Datasets

We perform our experiment on nine different CWS datasets. AS, CIT, MSR, and PKU datasets are from the SIGHAN2005 bakeoff (Emerson, 2005). CNC is from the Chinese corpus. CTB6 (XUE et al., 2005) is from the Chinese Penn Treebank. SXU dataset is from the SIGHAN2008 bakeoff (Jin and Chen, 2008). UD is from Zeman et al. (2017). ZX (Zhang et al., 2014) corpus is segmented from a novel called ZuXian. In Chinese Word Segmentation, the F1 score is used to evaluate the performance. The OOV recall is used to evaluate an MCCWS model's generalization ability. We report our F1 score and OOV recall on the test set according to the best checkpoint on the development set.

### 4.2 Experimental Setting

We preprocess all nine datasets by replacing consecutive English letters and numbers with 'a and '0'

respectively. The optimizer we used for finetuning is AdamW (Loshchilov and Hutter). Furthermore, their moving average coefficients are set to (0.9, 0.999). We set our learning rate to $2 \times 10^{-5}$ with a linear warmup and a linear decay. The warmup rate is 0.1 times the total training steps. Our model is finetuned for 5 epochs. We use the gradient accumulation with step 2 on a batch size 16 which approximates the effect of batch size 32. The value of $\alpha$ in Equation (6) is 0.3, and $\tau$ in Equation (5) is 0.1. We use the label smoothing technique on the word segmentation decoder, and the smoothing factor is set to 0.1. We refrain from using label smoothing on the criteria classifier because we aim for the model to precisely distinguish the differences between datasets. We run all our experiments on Intel Xeon Silver 4216 CPU and an Nvidia RTX 3090 GPU.

### 4.3 Main Results

#### 4.3.1 F1 score

Table 1 shows our F1 score on nine datasets. Our method achieves SoTA on 6 out of 9 datasets. We also report the average F1 score on 4 (denoted as Avg.4) and 6 (denoted as Avg.6) to compare with other methods that did not evaluate their model on all nine datasets. The model that is most similar to ours is proposed by Ke et al. (2020). By adding a sentence representation task, our MCCWS model's performance on Avg.4 and Avg.6 can improve 0.25% and 0.29%, respectively. Huang et al., 2020b used a private structure and CRF decoder, which means more parameters for each criterion. However, with a simpler architecture, our model performs better on Avg.6 and Avg.9.

#### 4.3.2 OOV recall

Out-of-Vocabulary (OOV) recall is a critical evaluation benchmark to measure an MCCWS model's ability to segment unseen words. Table 2 shows our

| MCCWS Models | AS | CIT | CNC | CTB6 | MSR | PKU | SXU | UD | ZX | Avg.4 | Avg.6 | Avg.9 |
|---|---|---|---|---|---|---|---|---|---|---|---|---|
| Gong et al., 2019 | 77.33 | 73.58 | - | 83.89 | 64.20 | 69.88 | 78.69 | - | - | 71.248 | 74.595 | - |
| Huang et al., 2020a | - | - | 59.44 | 88.02 | 81.75 | 82.35 | 85.73 | 91.40 | 82.51 | - | - | - |
| Qiu et al., 2020 | 76.39 | 86.91 | - | 87.00 | 78.92 | 78.91 | 85.08 | - | - | 80.283 | 82.202 | - |
| Ke et al., 2020 | 79.26 | 87.27 | - | 87.77 | 83.35 | 79.71 | 86.05 | - | - | 82.398 | 83.902 | - |
| Ke et al., 2021 | 80.89 | 90.66 | 61.90 | 89.21 | 83.03 | 80.90 | 85.98 | **93.59** | 87.33 | 83.870 | 85.112 | 83.721 |
| Ours | **81.27** | **91.83** | **66.28** | **91.95** | **89.20** | **83.77** | **88.00** | 93.54 | **88.23** | **86.518** | **87.670** | **86.007** |

Table 2: OOV recall (in percentage) on all nine CWS datasets. (We show the result of the significance test in Table 8.) Avg.4: Average among AS, CIT, MSR, and PKU; Avg.6: Average among AS, CIT, CTB6, MSR, PKU, and SXU; Avg.9: Average among nine datasets; Ours: Our method

OOV recall on nine datasets. Our method achieves SoTA on eight out of the nine datasets, showing that our approach can better learn to segment sequences according to the context instead of relying on the in-vocabulary information.

### 4.4 Ablation Study

To understand the influence of various loss functions on the performance of the model, we first remove the criterion classification loss. The F1 score drops slightly by 0.008% (See Table 3). This result shows that the criterion classification task helps the model distinguish the criteria, but an MC-CWS model can learn most of the criteria difference by itself. Second, we exclude the sentence representation loss. The F1 score drops 0.049% (See Table 3), six times greater than the reduction observed upon removing the criterion classification task alone. The OOV recall drops dramatically and shows our model can segment unseen words by referring to the context. We can conclude that learning semantic knowledge can further enhance performance. Finally, we remove both additional tasks; the F1 score drops even more (See Table 3). Therefore, these two additional tasks both improve performance. Notably, the sentence representation task plays a pivotal role in this enhancement, so we can conclude that learning semantic knowledge can further enhance CWS performance.

| Architecture | Avg F1 score | Avg OOV recall |
|---|---|---|
| Ours | 97.653 | 86.007 |
| w/o criterion classification | 97.645 **(-0.008)** | 85.826 **(-0.181)** |
| w/o sentence representation | 97.604 **(-0.049)** | 85.349 **(-0.658)** |
| w/o both | 97.592 **(-0.061)** | 85.423 **(-0.584)** |

Table 3: Ablation study. Where "w/o both" indicate removing criteria classification and sentence representation tasks

To prove that learning semantic knowledge enhances our model's segmentation ability, we tried MSE, cosine embedding, and loss function proposed by Zhang et al. (2022) to learn the sentence representation (See Table 4). MSE loss and cosine embedding loss are used to keep the two hidden representations we get from [CLS] token with different dropout masks similar. Additionally, we use the loss function revised by Zhang et al. (2022) to obtain better sentence embedding. No matter which method we use for learning sentence representation leads to improvement. The F1 score and OOV recall are better than without the sentence representation task. Therefore, we prove that learning representation helps the model segment input sentence better. In the end, our model and the gold label also show the same result as our analysis, but the model that is not trained by sentence representation task cannot deliver the correct result.

| Architecture | Avg F1 score | Avg OOV recall |
|---|---|---|
| Ours | 97.653 **(+0.049)** | 86.007 **(+0.658)** |
| MSE loss | 97.635 **(+0.031)** | 85.838 **(+0.489)** |
| Cosine Emb loss | 97.644 **(+0.040)** | 85.624 **(+0.275)** |
| Zhang et al., 2022 | 97.627 **(+0.023)** | 85.707 **(+0.358)** |
| w/o sentence representation | 97.604 | 85.349 |

Table 4: Performance of learning the sentence representation with different loss functions. (More details can find in Appendix B)

### 4.5 Case Study

We have selected a sentence from the **MSR** dataset and present the respective segmentations offered by different models in Table 5. The character sequence "证券登记结算公司" can be interpreted either as a single word or as a combination of individual components, namely that "证券 / 登记

| Source | Sentence |
|--------|----------|
| Original (input text) | 上海、深圳证券交易所及其下属证券登记结算公司 |
| Prediction (ours) and Gold | 上海 / 、 / 深圳证券交易所 / 及其 / 下属 / 证券 / 登记 / 结算 / 公司 
 (Interpreted by GPT-4 : Shanghai and Shenzhen Stock Exchanges and **their affiliated securities registration and clearing companies**) |
| Prediction (w/o sentence representation) | 上海 / 、 / 深圳证券交易所 / 及其 / 下属 / 证券登记结算公司 
 (Interpreted by GPT-4 : Shanghai and Shenzhen Stock Exchanges and **an affiliated company** that offers the services of securities registration and clearing) |

Table 5: The sentence is picked from the **MSR** dataset, and our model can deliver the correct segmentation. In contrast, the CWS model that did not train by the sentence representation task segmented the sentence with a wrong comprehension.

/ 结算 / 公司." While the character sequence remains unsegmented, the sequence's interpretation is "a company that offers the services of securities registration." In contrast, the segmented representation conveys the meaning of "securities registration and clearing companies." Based on the contextual cues, the semantic of the segmented representation must include all companies related to securities registration and clearing. Therefore, the segmented sequence is superior in meaning and interpretation. In the end, our model and the gold label also show the same result as our analysis, but the model that is not trained by sentence representation task cannot deliver the correct result.

## 5 Conclusion

In this paper, we introduce a novel training task to CWS, achieving the SoTA F1 score and OOV recall on several datasets. We then demonstrate that learning sentence representation is more crucial than adding criterion classification objective function because our model can learn the criteria differences during training in most cases. After realizing each context, our model can also deliver different results for a sequence of words with different contexts (More examples can be found in Table 6). Finally, because our model is simple and without any additional components, it can be easily utilized as a base model to finetune for specific datasets when only a single criterion is required.

## Acknowledgment

This work was funded by the National Science and Technology Council, Taiwan, 112-2223-E-006-009. Finally, thanks Jo-Ting, Chen for writing suggestion.

## Limitations

Several previous works did not release their code, so we did not reproduce their experiment. Ke et al., 2021; Huang et al., 2020a; Qiu et al., 2020; Huang et al., 2020b also face the same problem. We refer to previous comparison methods to compare our results with previous works and surpass their performance. Even though comparing the result without reproducing other work's experiments is slightly unfair, we still inevitably do so in Table 1 and Table 2.

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

## A Segment Examples

We list another case in Table 6 to show that our model can generate different segmentation results according to the context.

## B Loss for Learning Sentence Representation

In this section, we list all the loss functions we used for learning sentence representation. The MSE loss is shown as:

$$\mathcal{L}_{\text{MSE}} = \frac{1}{N} \sum_{i=1}^{N} (h_{0i} - h_{0i}^{+})^2. \tag{7}$$

The cosine embedding loss is shown as:

$$\mathcal{L}_{\text{cos}} = \frac{1}{N} \sum_{i=1}^{N} (1 - \text{sim}(h_{0i}, h_{0i}^{+})). \tag{8}$$

$N$ stands for the batch size. $h_{0i}$ and $h_{0i}^{+}$ are the hidden representation of the [CLS] token from the same input text with different dropout masks. $\text{sim}(h_{0i}, h_{0i}^{+})$ represents the cosine similarity and can be calculated as $\frac{h_{0i}^T h_{0i}^+}{||h_{0i}|| \cdot ||h_{0i}^+||}$. These two loss functions minimize the difference between the representation of the input text with different dropout masks and make the model realize these two input texts have the same meaning.

The third loss function is based on contrastive learning. It was proposed by Zhang et al. (2022) :

$$\mathcal{L}_{\text{ArcCSE}} = -\log \frac{e^{\cos(\theta_{i,i^+}+m)/\tau}}{e^{\cos(\theta_{i,i^+}+m)/\tau} + \sum_{j=1, i \neq j}^{N} e^{\cos(\theta_{i,j})/\tau}}, \tag{9}$$

where $m$ is viewed as a decision boundary between a positive pair and a negative pair. $\theta_{i,j}$ is the angle between two hidden representation $h_i$ and $h_j$. $\theta_{i,j}$ can be calculated as :

$$\theta_{i,j} = \arccos(\frac{h_i^T h_j}{||h_i|| \cdot ||h_j||}) \tag{10}$$

By adding a decision boundary, the model can distinguish positive pairs and negative pairs more precisely.

## C Significance Test

We use the t-test to demonstrate that our model's performance surpasses the previous SoTA and show the results in Table 7 and 8.

| Original Sentence | Segmentation Results from Our Model |
|---|---|
| 请小心使用,不要用坏了 | 请-小心-使用-,-不要-用坏-了 |
| 不要用坏了,别间厕所才可以正常使用 | 不要-用-坏-了-,-别-间-厕所-才-可以-正常-使用 |

Table 6: An example shows that our model can segment sentences according to the context. The hyphen "-" denotes segmentation.

| Experiments | Seed | AS | CIT | CNC | CTB6 | MSR | PKU | SXU | UD | ZX | Avg.9 |
|---|---|---|---|---|---|---|---|---|---|---|---|
| Ours | 927 | 96.81 | 98.16 | 97.41 | 97.93 | 98.39 | 96.94 | 97.82 | 98.30 | 97.12 | 97.653 |
| | 1564 | 96.74 | 98.17 | 97.44 | 97.89 | 98.34 | 96.97 | 97.79 | 98.31 | 97.09 | 97.637 |
| | 7849 | 96.76 | 98.09 | 97.42 | 97.93 | 98.37 | 96.93 | 97.71 | 98.39 | 97.07 | 97.630 |
| | 8453 | 96.72 | 98.09 | 97.45 | 97.98 | 98.45 | 96.99 | 97.74 | 98.30 | 97.04 | 97.639 |
| | 9416 | 96.75 | 98.25 | 97.44 | 97.86 | 98.37 | 96.97 | 97.70 | 98.22 | 97.09 | 97.628 |
| | Avg | 96.756 | 98.152 | 97.432 | 97.918 | 98.384 | 96.960 | 97.752 | 98.304 | 97.082 | 97.6374 |
| | Std | 0.030 | 0.059 | 0.014 | 0.041 | 0.037 | 0.022 | 0.046 | 0.054 | 0.026 | 0.0088 |

Table 7: F1 score of 5 different trials. Seed: Random seed we used in the experiment. **Avg.9**: Average among nine datasets. **Avg**: Average OOV recall among 5 trials. **Std**: Standard deviation among 5 trials. We use t-test to show that our method can produce better results than previous SoTA statistically. (Based on the result of our trails, we perform a one-tailed t-test with the hypothesis that our average F1 score $\leq$ the previous SoTA F1 score, and the calculated $t$-value is approximately 15.279. With a significance level of $\alpha = 0.05$ and degrees of freedom 4, the critical value is 2.776. Since the $t$-value exceeds the critical value, we reject the null hypothesis and conclude that our method is significantly better than the previous SoTA.)

| Experiments | Seed | AS | CIT | CNC | CTB6 | MSR | PKU | SXU | UD | ZX | Avg.9 |
|---|---|---|---|---|---|---|---|---|---|---|---|
| Ours | 927 | 81.27 | 91.83 | 66.28 | 91.95 | 89.20 | 83.77 | 88.00 | 93.54 | 88.23 | 86.007 |
| | 1564 | 80.36 | 91.83 | 66.20 | 91.63 | 88.04 | 83.77 | 87.33 | 93.81 | 87.29 | 85.584 |
| | 7849 | 80.44 | 91.61 | 65.95 | 91.51 | 88.24 | 83.75 | 87.03 | 94.35 | 87.38 | 85.584 |
| | 8453 | 80.11 | 91.35 | 66.10 | 91.54 | 88.00 | 84.34 | 87.86 | 94.08 | 86.88 | 85.584 |
| | 9416 | 80.42 | 92.37 | 66.32 | 91.34 | 88.41 | 83.36 | 87.15 | 93.34 | 87.96 | 85.630 |
| | Avg | 80.520 | 91.798 | 66.170 | 91.594 | 88.378 | 83.798 | 87.474 | 93.824 | 87.547 | 85.6778 |
| | Std | 0.393 | 0.336 | 0.133 | 0.201 | 0.437 | 0.313 | 0.387 | 0.363 | 0.485 | 0.1656 |

Table 8: OOV recall of 5 different trials. Seed: Random seed we used in the experiment. **Avg.9**: Average among nine datasets. **Avg**: Average OOV recall among 5 trials. **Std**: Standard deviation among 5 trials. We use t-test to show that our method can produce better results than previous SoTA statistically. (Based on the result of our trails, we perform a one-tailed t-test with the hypothesis that our average OOV recall $\leq$ the previous SoTA OOV recall, and the calculated $t$-value is approximately 23.638. With a significance level of $\alpha = 0.05$ and degrees of freedom 4, the critical value is 2.776. Since the $t$-value exceeds the critical value, we reject the null hypothesis, and conclude that our method is significantly better than the previous SoTA.)