# OpenReview forum: "Improving Multi-Criteria Chinese Word Segmentation through Learning Sentence Representation"
_EMNLP/2023/Conference — EMNLP 2023 Findings_

### Official Review · Reviewer_hrZu · 2023-07-25

**Soundness:** 4

**Excitement:**

3: Ambivalent: It has merits (e.g., it reports state-of-the-art results, the idea is nice), but there are key weaknesses (e.g., it describes incremental work), and it can significantly benefit from another round of revision. However, I won't object to accepting it if my co-reviewers champion it.

**Paper Topic And Main Contributions:**

This paper proposed a new method for multi-criteria Chinese word segmentation where segmentation crterions vary in different datasets. They propose to model the difference in criterions with a special token and criteria specific training objective. Moreover, they claim that the sentence meaning can help word segmentation, and introduce the contrastive objective in SIMCSE as an addtional objective. The main contribution of this paper would be a new idea and engineering product for multi-criteria CWS.

**Reasons To Accept:**

This paper proposed to consider criteria and sentence meaning when segmenting the words, which is reasonable and effective, bringing higher oov recall rate. The experiment results are comprehensive.

**Reasons To Reject:**

The absolute improvement for bringing criteria and sentence meaning is not significant, the method for modeling criteria and sentence meaning is rather simple. The average score on the popularly applied four datasets does not surpass SOTA.

**Reproducibility:**

5: Could easily reproduce the results.

**Reviewer Confidence:**

4: Quite sure. I tried to check the important points carefully. It's unlikely, though conceivable, that I missed something that should affect my ratings.

---

> ### Author Rebuttal · Authors · 2023-08-27
>
> Thank you for evaluating our paper and providing valuable insights for enhancement.
>
> **Discussion for the popular datasets**
>
> We can not only focus on the three datasets we failed to surpass the SoTA. We analyzed all the datasets and found some label inconsistencies.
>
> We first examined the difference between PKU’s training and testing sets to understand why our model underperformed compared to the SoTA. We selected 320 character sequences, corresponding to words in the training or test sets. These selected words are segmented differently in the two sets. Our model segmented these sequences with a 91.875% match to the training set. After addressing these label inconsistencies in the test set, our model’s performance has a 0.38% improvement in the F1 score and a 2.53% boost in OOV recall. Consequently, the PKU dataset requires modifications to maintain its validity in evaluating model performance.
>
> |          | F1-score | OOV-recall |
> | -------- | -------- | -------- |
> | Original performance | 96.94 | 83.77 |
> | Addressed label inconsistency | 97.32 **(+0.38)** | 86.30 **(+2.53)** |
>
> After the manual inspection of the PKU test set, we performed further automatic analysis with MCCWS to measure the label consistency of all the datasets. We treated the training and test sets from each dataset as different datasets with different criteria. For each time, we trained the MCCWS model with only the training set and the test set belonging to the dataset, with the criteria tokens as {dataset_name} for the training set and {dataset_name_**test**} for **the test set**. The trained MCCWS model was then tested on the test set of each dataset. In this experiment, if the training and test sets in a dataset share the same criteria, there should be no difference in the results inferred from the MCCWS model using different criteria tokens of each dataset.
> |Criteria token|AS|CIT|MSR|PKU|
> |-|-|-|-|-|
> |{dataset_name_test}|99.712|99.794|99.832|99.606|
> |{dataset_name}|98.904 **(-0.808)**|99.786 **(-0.008)**|99.826 **(-0.006)**|99.464 **(-0.142)**|
>
>
> From the table above, the AS and PKU datasets display higher scores of label inconsistency, which may cause unsatisfying performances of our method on these datasets. Therefore, it may be more appropriate to evaluate an approach across all CWS datasets instead of the four popular ones.
> We provide the results of the remaining datasets in the table below.
>
> |Criteria token|CNC|CTB6|SXU|UD|ZX|
> |-|-|-|-|-|-|
> |{dataset_name_test}|99.672|99.834|99.754|99.956|99.830
> |{dataset_name}|99.656 **(-0.016)**|99.862 **(+0.028)**|99.748 **(-0.006)**|99.938 **(-0.018)**|99.718 **(-0.112)**
>
> **Significance of improvement**
>
> We argue that the notable increase in OOV (Out-of-Vocabulary) recall of our method indicates a significant improvement over the CWS datasets. Additionally, while the absolute improvement in F1-score might not seem significant, it is essential to consider that the existing state-of-the-art methods have already pushed performance to a high level, with an average F1-score over 97%. Even a modest improvement can represent an important step forward.
>
> **Simple method**
>
> We argue that a simple but effective method is actually a strength of the proposed method.
> This work first addresses the context issue among the existing CWS methods, and we offer a simple and straightforward solution to alleviating the issue with sentence representation learning.
> Due to the simplicity of our method, it can be easily implemented and extended to future CWS research or applications.

---

### Official Review · Reviewer_9axb · 2023-08-05

**Soundness:** 3

**Excitement:**

3: Ambivalent: It has merits (e.g., it reports state-of-the-art results, the idea is nice), but there are key weaknesses (e.g., it describes incremental work), and it can significantly benefit from another round of revision. However, I won't object to accepting it if my co-reviewers champion it.

**Paper Topic And Main Contributions:**

Recent models on Chinese word segmentation tend to learn the segmentation knowledge through in-vocabulary words rather than understanding the meaning of the entire context.

To address the problem, the paper proposes a context-aware approach, in which unsupervised sentence representation learning is used in the multi-criteria training framework.

Experiments show that the proposed method achieves SoTA performance on F1 scores for six of the nine CWS benchmark datasets and OOV recalls for eight of nine.

Overall, the paper shows an empirical result on multi-criteria CWS jointly trained with unsupervised sentence representation learning, and achieve a small improvement on overall F1 score and larger improvements on OOV recall.

----
Update:
I raised the score according to the results in the rebuttal.
It is still not clear how context-aware information and joint task will help. Should give more evidence besides the score.

**Reasons To Accept:**

a) The paper is well written. The description of the model is clear.

b) The author conducts extensive experiments to verify the performance of the proposed model on different benchmarks.

**Reasons To Reject:**

1.	The paper looks incremental, as the only improvement on the model is to introduce the simCSE loss compared with previous multi-criteria CWS model (Ke et al. 2020). And the improvement on F1 score is small.

2.	The previous model based on BERT also includes [CLS] token, though not specially trained with simCSE, can also be regarded as the use of entire context. A relevant baseline should be a two stage-model (first train simCSE on BERT, and then train multi-criteria CWS model) to see if joint training can help rather than the improvement of the base model.

**Reproducibility:**

4: Could mostly reproduce the results, but there may be some variation because of sample variance or minor variations in their interpretation of the protocol or method.

**Reviewer Confidence:**

3: Pretty sure, but there's a chance I missed something. Although I have a good feel for this area in general, I did not carefully check the paper's details, e.g., the math, experimental design, or novelty.

---

> ### Author Rebuttal · Authors · 2023-08-27
>
> Thank you for evaluating our paper and providing valuable insights for enhancement.
>
> **Contribution on context-awareness**
>
> Our paper not only introduces the SimCSE loss but also brings in a context-aware approach to CWS. While previous models might have included a [CLS] token, our approach specifically trains the model to leverage the contextual meaning of the entire sentence, which we believe is a conceptual step forward. Furthermore, we would like to emphasize that we tried base and near SoTA sentence representation methods to learn the meaning of the input text (Section 4.4).
>
>
> **Contribution on performance improvements**
>
> We argue that the notable increase in OOV (Out-of-Vocabulary) recall is a significant contribution.
> OOV is a common challenge in Chinese Word Segmentation (CWS), and improving OOV recall can be more meaningful than just optimizing F1 scores. Moreover, with respect to F1 scores, our method remains an effective approach across multiple CWS benchmark datasets, whether evaluated on an average of six or all nine datasets. As a result, our work presents notable contributions, which we believe will be valuable for future CWS research.
>
> **Comparison with a two-stage baseline**
> We have conducted an additional experiment that implements a two-stage training approach: first training SimCSE on BERT, followed by multi-criteria CWS. The result are shown in the following table for the average of all nine datasets:
>
>
> |          | F1-score | OOV-recall |
> | -------- | -------- | -------- |
> | Joint training (proposed) | 97.65     | 86.007     |
> | Two-stage | 97.62 **(-0.03)**     | 85.610 **(-0.397)**   |
> | w/o sentence representation | 97.60 **(-0.05)**     | 85.349 **(-0.658)**   |
>
> This result indicates that the proposed joint training method outperforms the two-stage training approach and the one without the sentence representation objective. We will also include this result in our camera-ready version.

---

### Official Review · Reviewer_kPXV · 2023-08-07

**Soundness:** 4

**Excitement:**

3: Ambivalent: It has merits (e.g., it reports state-of-the-art results, the idea is nice), but there are key weaknesses (e.g., it describes incremental work), and it can significantly benefit from another round of revision. However, I won't object to accepting it if my co-reviewers champion it.

**Paper Topic And Main Contributions:**

A new training loss for improving Chinese word segmentation


This paper introduces a new training loss for Chinese word segmentation. That is, it incorporates the contrastive objective for learning unsupervised sentence representation. The idea is to encourage the understanding of the entire sentence. Experiments show marginal improvements in overall performance and moderate improvements in OOV recall.

**Reasons To Accept:**

1. The proposed method is simple and universal (reaches state-of-the-art (SoTA) performance on F1 scores for six of the nine CWS benchmark datasets)

**Reasons To Reject:**

1. This work seem incremental in that it combines CWS and recent unsupervised sentence representation learning.
2. This overall performance improvements seem marginal.

**Reproducibility:**

3: Could reproduce the results with some difficulty. The settings of parameters are underspecified or subjectively determined; the training/evaluation data are not widely available.

**Reviewer Confidence:**

4: Quite sure. I tried to check the important points carefully. It's unlikely, though conceivable, that I missed something that should affect my ratings.

---

> ### Author Rebuttal · Authors · 2023-08-27
>
> Thank you for evaluating our paper and providing valuable insights for enhancement.
>
> For the first issue, it may seem our approach simply adds an extra training task to the CWS model. However, let us consider an example in the table to emphasize the significance of the model comprehending the entire sentence for segmentation. The sentence is picked from the MSR dataset, and our model can deliver the correct segmentation. In contrast, the CWS model that did not train by the sentence representation task segmented the sentence with a wrong comprehension. The result implies that a CWS model might give out inadequate segmentation without sufficient understanding of a sentence. We will provide more examples in the camera-ready version.
>
> | Original | 上海、深圳证券交易所及其下属证券登记结算公司 |
> | -------- | -------- |
> | Gold     | 上海 / 、 / 深圳证券交易所 / 及其 / 下属 / 证券 / 登记 / 结算 / 公司 (Interpreted by GPT-4 : Shanghai and Shenzhen Stock Exchanges and **their affiliated securities registration and clearing companies**)     |
> | Prediction (Previous) | 上海 / 、 / 深圳证券交易所 / 及其 / 下属 / 证券登记结算公司 (Interpreted by GPT-4 : Shanghai and Shenzhen Stock Exchanges and **an affiliated company** that offers the services of securities registration and clearing)|
>
>
> For the second issue, we knew that the absolute improvements in the F1 score seemed marginal, so we manually analyzed the cases in which our model made mistakes. We discovered that most of the mistakes are OOV or label inconsistency. To realize how the label inconsistency problem degrades our model's performance, we tried to correct **320 words'** labels in the PKU dataset according to its training set's labeling rules. The performance on the F1 score can improve by 0.38%. However, we can not use the dataset we revised. Otherwise, it would not be a fair comparison.  In conclusion, the label inconsistency between the training and testing sets is a big problem in the CWS field.

---

### Meta-Review · Area_Chair_sUe6 · 2023-09-17

**Recommendation:** 3

**Metareview:**

This paper describes a new approach to multi-criteria Chinese word segmentation (MCCWS) that improves context awareness by incorporating a secondary objective of learning sentence representations. The authors report that their approach achieves state of the art F1 and OOV recall on many of the available CWS benchmark datasets.

The reviewers agree that the proposed approach is well motivated, simple, and mostly sound, and they note that the set of experiments carried out is comprehensive and thorough. The reviewers were less enthusiastic, however, about the overall contribution of the work. In particular, they note that the reported improvements over the prior s.o.t.a. are minimal, and the proposed approach fails to yield improvements for some of the datasets. One reviewer did raise their soundness score after the rebuttal in response to additional results provided by the authors, but they remained ambivalent about the significance of the work.

---

### Decision · Program_Chairs · 2023-10-07

**Decision:**

Accept-Findings

**Comment:**

This paper describes a new approach to multi-criteria Chinese word segmentation (MCCWS) that improves context awareness by incorporating a secondary objective of learning sentence representations. The authors report that their approach achieves state of the art F1 and OOV recall on many of the available CWS benchmark datasets.

The reviewers agree that the proposed approach is well motivated, simple, and mostly sound, and they note that the set of experiments carried out is comprehensive and thorough. The reviewers were less enthusiastic, however, about the overall contribution of the work. In particular, they note that the reported improvements over the prior s.o.t.a. are minimal, and the proposed approach fails to yield improvements for some of the datasets. One reviewer did raise their soundness score after the rebuttal in response to additional results provided by the authors, but they remained ambivalent about the significance of the work.